# Causality-attended Representation Disentanglement with Structural Alignment for Fairness-aware Graph Adaptation

## Abstract

This paper studies the problem of fairness-aware graph adaptation, which aims to transfer knowledge from a labeled source graph to an unlabeled target graph with the consideration of fairness. Previous approaches usually utilize adversarial learning to learn invariant graph representations of sensitive attributes. However, these approaches assume that sensitive attributes are known on the target graph, which may not always be the case in the real world. Towards this end, we propose a new approach named Causality-attended Representation Disentanglement with Structural Alignment (COSTA) for fairness-aware graph adaptation. The core of our COSTA is to build a causal graph to guide the feature representation disentangle with enhanced fairness. In particular, our causal graph explores the underlying mechanism of graph generation and then utilizes a sensitive encoder and a causal encoder for feature extraction. To ensure representation disentanglement, we minimize the mutual information between causal representations and sensitive representations, considering the conditional distribution. To make use of unlabeled data, we generate pseudo-labels for both target and sensitive attributes and measure the similarity relations for unbiased node representations. To further mitigate the domain shift, we construct a fairness-aware bipartite graph, which can further guide the domain alignment. Extensive experiments on benchmark datasets validate the effectiveness of the proposed method in comparison to competing baselines.

## 1 Introduction

Graph Neural Networks (GNNs)' social impact has broadened significantly since their adoption as the *de facto* approach for modeling complex relational structures Kipf & Welling (2017); Xu et al. (2019). Thanks to their powerful ability to capture both node features and relational dependencies through message passing mechanisms, GNNs have achieved remarkable success in numerous graph-based tasks Wu et al. (2020); Ju et al. (2024). Among these, node classification endeavors to predict the label of each node in a graph and stands as one of the most fundamental tasks in numerous graph-learning-based applications, i.e., community detection Bianchi et al. (2020), cross-model retrieval Qian et al. (2022) and molecular property prediction Wang et al. (2021). Nevertheless, graph learning often suffer from fairness issues due to inherent biases in graph data, which can be further exacerbated by the message-passing mechanisms of GNNs Chen et al. (2024).

To mitigate this issue, recent years have witnessed considerable efforts toward fair graph learning. Depending on the stage at which the fairness intervention is introduced, existing approaches are generally categorized into pre-, in-, and post-processing strategies. Pre-processing strategies aim to mitigate bias prior to the model training phase through manipulating the original input graph, i.e., node feature masking Köse & Shen (2021), structure rewiring Spinelli et al. (2021); Dong et al. (2022). By contrast, in-processing strategies incorporate fairness constraints to modify the objective function during the training process, such as through regularization Agarwal et al. (2021); Jiang et al. (2024), adversarial debiasing Ling et al. (2023), and disentangled representation learning Zhu et al. (2024a); Lee et al. (2025). Finally, post-processing strategies adjust the model's outputs after training to mitigate bias and enhance fairness Dai & Wang (2021); Zhang et al. (2024b).

However, most of these approaches assume access to sensitive attributes (e.g., race, gender) for all nodes during training, which is often impractical due to privacy concerns or missing demographic information. Graph transfer learning seeks to leverage knowledge from a source graph where such attributes are available to improve performance on a target graph with scarce or missing labels Han et al. (2021); Zhu et al. (2021; 2024b). Within this paradigm, graph domain adaptation (GDA) has emerged as a key strategy that directly aligns the distributions of source and target graphs in the learned GNN-representation space, and there are numerous efforts that have been proposed recently Qiao et al. (2023); You et al. (2023); Liu et al. (2024). Therefore, this naturally spurs a question: *How can fairness knowledge be effectively transferred across graphs when sensitive labels are unavailable in the target domain?* This insight motivates the formalization of an innovative application scenario, namely, graph fairness adaptation.

Nonetheless, developing such a graph fairness adaptation framework for node classification remains a non-trivial task since it must address two basic challenges: ❶ *Sensitive Group Distribution Shift under Domain Discrepancy.* The structural heterogeneity and semantic shifts between source and target graphs implicitly alter the underlying distribution of sensitive groups, which destabilizes fairness knowledge transfer from the source and consequently weakens bias mitigation in the target domain. For example, prior studies have observed that a fair income prediction model developed in one state may lose its fairness when transferred to another state Ding et al. (2021). ❷ *Alignment Collapse during Entangled Information Transfer.* Existing methods typically rely on distribution discrepancy minimization Wu et al. (2023) and adversarial learning Dai et al. (2022) to align domain distributions, which inevitably induce unintended correlations between the target-relevant and sensitive-related information. Transferring such entangled information may create a conflict between fairness and performance objectives, potentially leading to distribution alignment collapse.

Towards this end, in this paper, we propose a **G**roup-acqui**R**ed Bipartit**E** **A**lignment framework (termed COSTA) for graph fairness adaptation, which aims to transfer the fairness knowledge from the source graph to the unlabeled target graph. Specifically, our COSTA incorporates a dual graph encoder with a two-fold mutual information (MI) constraint, enabling the model to disentangle task-relevant and sensitive-related representations. Based on this, we generate pseudo-sensitive labels for target graph nodes and partition them into corresponding demographic groups. To enhance fairness in pseudo-labeling, a group-acquired unbiased learning strategy is then introduced, which explicitly emphasizes negative pairs sharing identical sensitive labels. Finally, for each target graph node, we retrieve source nodes with the same target while distinct sensitive labels for bipartite graph construction between two domains and employ a bipartite-aware domain alignment to decorrelate sensitive and target information for enhanced fairness adaptation.

In a nutshell, the contributions of the paper are as follows: ❶ *New Perspective.* We highlight the limited or unavailable nature of sensitive information in graph fairness learning and introduce an underexplored yet graph fairness adaptation problem. To the best of our knowledge, this is the first attempt to explore this problem. ❷ *Novel Methodology.* We propose a novel framework termed COSTA, which generalizes both target and sensitive representations with a dual graph encoder under a two-fold MI constraint and performs group-acquired bipartite alignment for unbiased domain alignment. ❸ *Extensive Experiments.* We conduct extensive experiments on multiple benchmark datasets to evaluate COSTA. The results demonstrate that our framework achieves superior performance and fairness for the adaptation task. The code is available at `https://anonymous.4open.science/r/COSTA_ICLR_2026-EFBE/`.

## 2 PRELIMINARIES & PROBLEM DEFINITION

**Notations.** Let the *source domain graph* be denoted as $\mathcal{G}^{so} = \{\mathcal{V}^{so}, \mathcal{E}^{so}, \boldsymbol{X}^{so}, \boldsymbol{Y}^{so}, \boldsymbol{S}^{so}\}$, where $\mathcal{V}^{so}$ and $\mathcal{E}^{so}$ represents the node and edge set respectively. We use the adjacency matrix $\boldsymbol{A}^{so}$ to describe the structure information of the source domain graph, where $\boldsymbol{A}^{so}_{uv} = 1$ if there is an edge $(u, v) \in \mathcal{E}^{so}$, otherwise $\boldsymbol{A}^{so}_{uv} = 0$. The node feature matrix is given by $\boldsymbol{X}^{so} \in \mathbb{R}^{|\mathcal{V}^{so}| \times d}$, where each row $\boldsymbol{x}_v \in \mathbb{R}^d$ represents the $d$-dimensional feature vector of node $v$. The node sensitive attribute are specified as $\boldsymbol{S}^{so} = \{s_1, \ldots, s_{|\mathcal{V}^{so}|}\} \in \{0, 1\}^{|\mathcal{V}^{so}|}$, where $s_v$ is the sensitive label of node $v$. We consider the binary node classfication and the target node label matrix can be $\boldsymbol{Y}^{so} = \{y_1, \ldots, y_{|\mathcal{V}_{so}|}\} \in \{0, 1\}^{|\mathcal{V}^{so}|}$. Similarly, the *target domain graph* is denoted as $\mathcal{G}^{ta} = \{\mathcal{V}^{ta}, \mathcal{E}^{ta}, \boldsymbol{X}^{ta}\}$ with completely

unlabeled node set $\mathcal{V}^{ta}$ and edge set $\mathcal{E}^{ta}$. Note that to facilitate alignment, we construct a unified feature space across the source and target domain graphs.

**Definition 2.1** (**Sensitive Group**). The sensitive group of the source and target domain graph is partitioned by nodes according to their sensitive attribute, formally defined as:

$$\mathcal{V}_s^* = \{v \in \mathcal{V}^* | s_v = s\}, * \in \{so, ta\}. \tag{1}$$

**Definition 2.2** (**EO Group**). The Equality Odds (EO) group of the graph is formed by partitioning nodes according to both target label $y$ and sensitive attribute $s$ of the node:

$$\mathcal{V}_{y,s}^* = \{v \in \mathcal{V}^* | (s_v = s) \cap (y_v = y)\} * \in \{so, ta\}. \tag{2}$$

**Definition 2.3** (**Demographic Parity**). Demographic parity Calders et al. (2009) is achieved to ensure fairness by enforcing that nodes from different demographic groups have equal probabilities of being assigned positive predictions. Accordingly, $\Delta_{DP}$ of target domain graph can be:

$$\Delta_{DP} = |\mathbb{E}_{u \in \mathcal{V}^{ta}}(\hat{y}_u = 1 | s_u = 1) - \mathbb{E}_{v \in \mathcal{V}^{ta}}(\hat{y}_v = 1 | s_v = 0)|, \tag{3}$$

where $\hat{y}_v$ and $y_v$ denote the predicted and ground-truth label of node $v$. Consequently, the dependence between predictions $\hat{y}$ and sensitive attribute $s$, namely $\hat{y} \perp\!\!\!\perp s$, is measured by $\Delta DP$.

**Definition 2.4** (**Equalized Odds**). Equal odds Hardt et al. (2016) stipulates fairness by ensuring that the True Positive Rate (TPR) and False Positive Rate (FPR) are identical across demographic groups. Formally, $\Delta_{EO}$ of target domain graph is defined as:

$$\Delta_{EO} = \frac{1}{2} \sum_{y=0}^{1} |\mathbb{E}_{u \in \mathcal{V}^{ta}}(\hat{y}_u = y | y_u = y, s_u = 1) - \mathbb{E}_{v \in \mathcal{V}^{ta}}(\hat{y}_v = y | y_v = y, s_v = 0)|. \tag{4}$$

Note that $\Delta_{EO}$ quantifies the conditional independence between the predicted label $\hat{y}$ and sensitive attribute $s$ given the ground-truth label $y$, i.e., $\hat{y} \perp\!\!\!\perp s \mid y$.

**Problem Definition.** Graph fairness adaptation aims to transfer the target and sensitive knowledge from a labeled source domain graph to a completely unlabeled target domain graph. Specifically, given the labeled source domain graph $\mathcal{G}^{so}$ and the unlabeled target domain graph $\mathcal{G}^{ta}$ with the covariate shift assumption Ben-David et al. (2006; 2010), i.e., $\mathbb{P}(\mathcal{G}_{so}) \neq \mathbb{P}(\mathcal{G}_{ta})$ and $\mathbb{P}(\boldsymbol{Y}|\mathcal{G}_{so}) = \mathbb{P}(\boldsymbol{Y}|\mathcal{G}_{ta})$, the objective of graph fairness adaptation is to label the nodes within the target domain graph $\mathcal{G}^{ta}$ while ensuring both high performance and fairness.

## 3 THE PROPOSED COSTA

### 3.1 FRAMEWORK OVERVIEW

In this section, we present COSTA, a framework that achieves unbiased adaptation by disentangling target and sensitive feature transfer through group-acquired bipartite alignment. Figure 1 provides an overview of the framework and we present the details of each component below.

### 3.2 CAUSALITY-ATTENDED REPRESENTATION DISENTANGLEMENT WITH MUTUAL INFORMATION OPTIMIZATION

**Causal Graph Construction.** Since the key challenge of the graph fairness adaptation lies in identifying stable sensitive-free features that preserve fair prediction while suppressing sensitive-aware features across domains, we perform feature disentanglement based on the constructed causal graph. We formalize the dependencies between variables through a Structural Causal Model (SCM) Pearl et al. (2016), where the three key mechanisms can be defined as:

- Domain Latent Factorization: $C^* \leftarrow \mathcal{D}^*, * \in \{so, ta\} \rightarrow S^*$ ensures that task-relevant factor $C^*$ is preserved, while permitting residual dependencies attributed to sensitive factors $S^*$ across the source domain $\mathcal{D}^{so}$ and target domain $\mathcal{D}^{ta}$.
- Graph Generation: $C^* \rightarrow \mathcal{G}^* \leftarrow S^*$ specifies that the observed graph data of two domains $\mathcal{G}^*$ is generated through the causal variable and the bias fairness-aware variable.

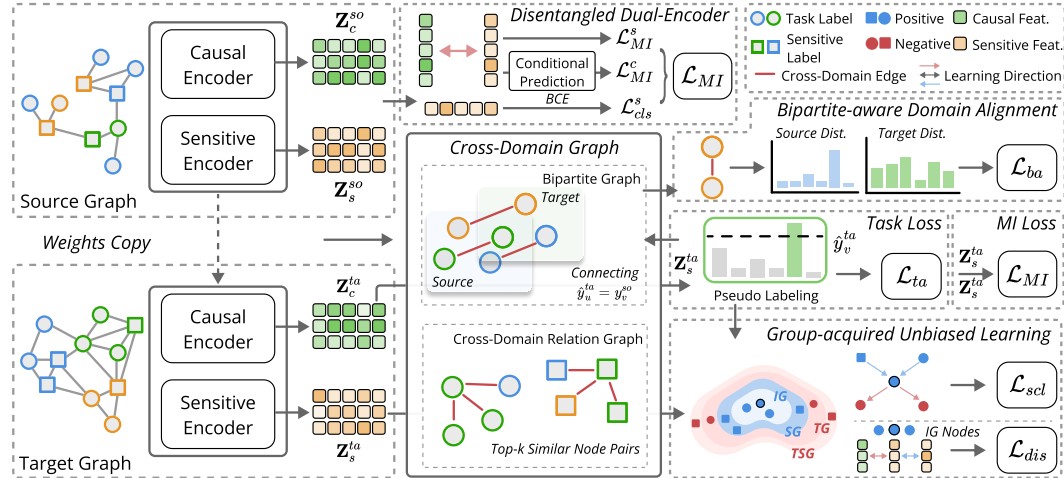

Figure 1: Illustration of COSTA with three modules: (1) Causality-attended Disentanglement: MI-based encoders disentangle task-relevant and sensitive factors. (2) Group-attended Unbiased Learning: Generate pseudo labels to support group-attended unbiased learning (3) Bipartite Domain Alignment: Reduce domain shift through clustering-based bipartite graph alignment

- Label Determination: $C^* \to Y^*$ indicates that the causal variable is the only endogenous parent to determine the ground-truth task label $Y^*$ under distribution shift.

Note that the spurious correlations of $C^*$ and $S^*$ within and between the graphs lead to poor fairness generalization under distribution shifts.

**Feature Disentanglement for Fairness Preservation.** Following the causal theory Wu et al. (2022), there exist a directed link between the variable $Y$ to its parent $PA(Y)$ in an SCM, if and only if a causal mechanism $Y = H(PA(Y), \epsilon_Y)$ persists, where the $\epsilon_Y \perp\!\!\!\perp PA(Y)$ is the exogenous noise of $Y$. In our setting, the mechanism can be represented as:

$$Y = H(PA(Y), S), Y \perp\!\!\!\perp S | C. \tag{5}$$

Thus, we disentangle $S$ and $C$ to preserve the causal effect of $C$ on $Y$, while eliminating the influence of $S$ for graph fairness adaptation. Specifically, we employ a GNN to obtain the node embeddings $\boldsymbol{Z}_s^{so}$ of the source domain graph and incorporate a sensitive discriminator $\xi(\cdot) : \boldsymbol{Z}_s^{so} \to \boldsymbol{S}$ to explicitly correlate with the sensitive attribute. The classification loss can be defined as:

$$\mathcal{L}_{cls}^s = - \sum_{v \in \mathcal{V}^{so}} \mathrm{BCE}(s_v, \xi(\boldsymbol{z}_{v,s}^{so})). \tag{6}$$

Based on the sensitive-aware embeddings, we employ another GNN to generate the causal task-relevant embedding $\boldsymbol{Z}_c^{so}$ and implement the graph fairness learning through:

$$\max \underbrace{I(\boldsymbol{Z}_c^{so}; \mathcal{D}^{so} | \boldsymbol{Z}_s^{so})}_{\text{Conditional Fair Prediction}} - \underbrace{\beta I(\boldsymbol{Z}_s^{so}; \boldsymbol{Z}_c^{so})}_{\text{Sensitive-Free}}, \tag{7}$$

where $I(\cdot, \cdot)$ denotes the mutual information. In practice, we derive a sample-based MI upper bound based on the Contrastive Log-ratio Upper Bound (CLUB) Cheng et al. (2020) for the aforementioned sensitive-free constraints, which can be formulated as:

$$\min I(\boldsymbol{Z}_s^{so}; \boldsymbol{Z}_c^{so}) := \mathcal{L}_{MI}^s = \frac{1}{M} \sum_{v=1}^M \left[ \log q_\theta(\boldsymbol{z}_{v,s}^{so} | \boldsymbol{z}_{v,c}^{so}) - \frac{1}{M} \sum_{u=1}^M q_\theta(\boldsymbol{z}_{u,s}^{so} | \boldsymbol{z}_{v,c}^{so}) \right], \tag{8}$$

where we leverage a Multilayer Perceptron (MLP) $q_\theta(\boldsymbol{z}_{v,s}^{so} | \boldsymbol{z}_{v,c}^{so})$ to approximate the conditional probability. For the requirement of the conditional fair prediction constraint, we leverage a model pre-trained with the target task label to generate the low-rank node embedding $\boldsymbol{\mu}_v$ given the high-dimension and sparsity of the source domain graph. Then, we employ a Conditional InfoNCE Gupta et al. (2021) for the MI lower-bound, which can be defined as:

$$\max I(\boldsymbol{Z}_c^{so}; \mathcal{D}^{so} | \boldsymbol{Z}_s^{so}) := \mathcal{L}_{MI}^c = \mathbb{E} \left[ \log \frac{\exp f(\boldsymbol{\mu}_v, \boldsymbol{z}_{v,c}^{so}, \boldsymbol{z}_{v,s}^{so})}{\frac{1}{M} \sum_{u=1}^M \exp f(\boldsymbol{\mu}_u, \boldsymbol{z}_{v,c}^{so}, \boldsymbol{z}_{v,s}^{so})} \right], \tag{9}$$

where $\boldsymbol{\mu}_v \sim p(\boldsymbol{\mu}_v | \boldsymbol{z}_{v,s}^{so})$ and $f(\cdot)$ denotes the conditional score function, where we implement as the weighted cosine similarity following the prior work Zhao et al. (2023):

$$f(\boldsymbol{\mu}_v, \boldsymbol{z}_{v,c}^{so}, \boldsymbol{z}_{v,s}^{so}) = \text{sim}(\boldsymbol{\mu}_v, \boldsymbol{z}_{v,c}^{so} + \alpha \boldsymbol{z}_{v,s}^{so}), \tag{10}$$

where $\alpha$ is the trade-off hyper-parameter and $\text{sim}(\cdot)$ is the cosine similarity function. Note that since the bias direction of the sensitive attribute, we implement the conditional sampling based on the direction function, namely $\{\boldsymbol{\mu}_{v'} | \pi(\boldsymbol{z}_{v,c}^{so}, \boldsymbol{z}_{v',c}^{so}) > 0\}$. So in this way, we can ensure that the sampled embedding belongs to the same bias direction w.r.t. the conditional distribution. The composite fairness learning objective for the source domain graph is $\mathcal{L}_{MI} = \mathcal{L}_{cls}^s + \mathcal{L}_{MI}^s - \mathcal{L}_{MI}^c$.

### 3.3 GROUP-ATTENDED PSEUDO-LABELING FOR UNBIASED REPRESENTATION LEARNING

To mitigate the label scarcity of the target domain graph while preventing fairness in pseudo-labels, we develop a fairness-preserving discriminative learning mechanism with group-acquired enhancement. Specifically, we employ the dual-encoder to get the node embedding $\boldsymbol{Z}_c^{ta}$ and $\boldsymbol{Z}_s^{ta}$ of the target domain graph and leverage them to determine the corresponding pseudo-label distribution,

$$\hat{y}_v^{ta} = \arg\max \psi(\boldsymbol{z}_{v,c}^{ta}), \ \hat{s}_v^{ta} = \arg\max \xi(\boldsymbol{z}_{v,s}^{ta}), \tag{11}$$

where $\psi(\cdot)$ projects the causal feature to the task label space. To preserve fairness in pseudo-label predictions, we decompose node similarities in the target domain graph into fine-grained types w.r.t. demographic groups, and explicitly penalize the model when it captures spurious correlations or sensitive information Park et al. (2022); Zhang et al. (2024a). Specifically, given an anchor node $v$, we categorize its similarity relations into three groups defined as follows.

- *Intra-Group* (**IG**): The similarity is determined according to the EO group, and the corresponding node group is formulated as $\mathcal{V}_{ig}^{ta}(v) = \{ig \in \mathcal{V}^{ta} | s_{ig} = \hat{s}_v^{ta} \cap y_{ig} = \hat{y}_v^{ta}\}$.
- *Sensitive Inter-Group* (**SG**): We define the similarity between an anchor and nodes that belong to the same target class but distinct sensitive attributes, with the corresponding node group formally denoted as $\mathcal{V}_{sg}^{ta}(v) = \{sg \in \mathcal{V}^{ta} | s_{sg} \neq \hat{s}_v^{ta} \cap y_{sg} = \hat{y}_v^{ta}\}$.
- *Target Inter-Group* (**TG**): The similarity is defined between an anchor and nodes that share the same sensitive attribute, while differing in target class, and the node group can be defined as $\mathcal{V}_{tg}^{ta}(v) = \{tg \in \mathcal{V}^{ta} | s_{tg} = \hat{s}_v^{ta} \cap y_{tg} \neq \hat{y}_v^{ta}\}$.

Building upon this, we encourage higher similarity among nodes within *IG* and *SG* than among those within *TG* to promote fairness in the target domain graph, formulated as:

$$\mathcal{L}_{scl} = -\frac{1}{|\mathcal{V}^{ta}|} \sum_{v \in \mathcal{V}^{ta}} \frac{1}{|\mathcal{V}_c(v)|} \sum_{u \in \mathcal{V}_c(v)} \log \frac{\phi_{u,c}}{\sum_{tg \in \mathcal{V}_{tg}^{ta}(v)} \phi_{tg,c}}, \tag{12}$$

where $\mathcal{V}_c(v) = \mathcal{V}_{ig}^{ta}(v) \cup \mathcal{V}_{sg}^{ta}(v)$ and $\phi_{*,c} = \exp(\boldsymbol{z}_{v,c}^{ta} \cdot \boldsymbol{z}_{*,c}^{ta}/\tau), * \in \{u, tg\}$. Meanwhile, we further decorrelate the sensitive information within *IG* of the target domain graph, defined as:

$$\mathcal{L}_{dis} = -\frac{1}{|\mathcal{V}^{ta}|} \sum_{v \in \mathcal{V}^{ta}} \frac{1}{|\mathcal{V}_s(v)|} \sum_{u \in \mathcal{V}_{ig}^{ta}(v)} \log \frac{\phi_{u,s}}{\sum_{ig \in \mathcal{V}_{ig}^{ta}(v)} \phi_{ig,c}}, \tag{13}$$

where $\phi_{u,s} = \exp(\boldsymbol{z}_{v,c}^{ta} \cdot \boldsymbol{z}_{u,s}^{ta}/\tau)$. The group-acquired unbiased loss is $\mathcal{L}_{ub} = \mathcal{L}_{scl} + \mathcal{L}_{dis}$, which emphasizes negative pairs with the same sensitive labels to promote fairness under distribution shift.

### 3.4 DOMAIN ALIGNMENT WITH FAIRNESS-AWARE BIPARTITE

Despite the pseudo-labels are generated for the target domain graph, the severe shift between source and target domains persists, which may lead to unreliable supervision signals. To mitigate this, we construct a bipartite graph where edges connect similar sensitive-free node pairs across domains and introduce a bipartite-aware mechanism for the domain alignment. Specifically, given the node $u \in \mathcal{V}^{ta}$ from target domain graph with pseudo-labels $\hat{y}_u^{ta}$ and $\hat{s}_u^{ta}$, we retrieve nodes $v \in \mathcal{V}^{so}$ from source domain graph and add edges between pair $(u, v)$ as:

$$\boldsymbol{B}_{uv} = \begin{cases} 1, & \hat{y}_u^{ta} = y_v^{so} \cap \hat{s}_u^{ta} \neq s_v^{so} \\ 0, & \text{otherwise} \end{cases}, \tag{14}$$

where $\boldsymbol{B}$ denotes the adjacency matrix of the bipartite graph, with each source and target domain sample being a node. To facilitate domain alignment, we impose a graph clustering constraint on the label predictions, enforcing that the majority of connected edges lie within the same clusters, which not only reduces domain discrepancy but also strengthens the discriminative capacity of task-relevant embeddings. The bipartite alignment loss can be defined as:

$$\mathcal{L}_{ba} = \|(\boldsymbol{I} - \boldsymbol{L}) - \boldsymbol{P}\boldsymbol{P}^T\|_F^2, \tag{15}$$

where $\boldsymbol{P}$ is the label prediction matrix, which is constructed as $\boldsymbol{P} = [\mathbb{1}(\boldsymbol{Y}^{so}), \xi(\boldsymbol{Z}_c^{ta})]$ with $\mathbb{1}[\cdot]$ as the one-hot function. $\boldsymbol{L}$ denotes the normalized Laplacian matrix of $\boldsymbol{B}$, and $\boldsymbol{I}$ is the identity matrix. Note that we ignore the intra-domain relationships and the above loss can be rewritten as:

$$\mathcal{L}_{ba} = -2\sum_{u,v} \frac{\boldsymbol{B}_{uv}}{\sqrt{d_u}\sqrt{d_v}} \boldsymbol{p}_u^T\boldsymbol{p}_v + \sum_{u,v} \boldsymbol{B}_{uv}(\boldsymbol{p}_u^T\boldsymbol{p}_v)^2 + \text{const}, \tag{16}$$

where $d_u$ denotes the degree of node $u$ in the bipartite graph.

### 3.5 Overall Optimization

To alleviate label scarcity while avoiding overconfidence in target domain pseudo-labels, we also quantify the prediction certainty through maximum class probability:

$$m_v^{ta} = \max_{k'} \psi(\boldsymbol{z}_{v,c}^{ta})[k], \tag{17}$$

where $m_v^{ta}$ is the confidence score. Then, we introduce an adaptive confidence score $\tau_k$ for class $k$ based on the estimated prediction certainty, defined as

$$\tau_k = \mathcal{M}_k \cdot \tau, \ \mathcal{M}_k = \max\{m_v^{ta} | \arg\max_{k'} \psi(\boldsymbol{z}_{v,c}^{ta})[k'] = k\} \tag{18}$$

where $\tau$ denotes the threshold. And the confident set $\mathcal{C}$ of the target domain graph can be refined as:

$$\mathcal{C} = \{v | v \in \mathcal{V}^{ta}, k = \arg\max_{k'} \psi(\boldsymbol{z}_{v,c}^{ta})[k'], m_v^{ta} > \tau_k\}. \tag{19}$$

We further optimize the model within the confident set $\mathcal{C}$ for cross-domain stability:

$$\mathcal{L}_{ta} = -\frac{1}{|\mathcal{C}|} \sum_{v \in \mathcal{C}} \log \psi(\boldsymbol{z}_{v,c}^{ta})[\hat{y}_v^{ta}]. \tag{20}$$

Similarly, we also leverage a threshold $\delta$ to filter out nodes with a high-confidence sensitive label for group-acquired enhancement. The overall objective of our graph fairness adaptation framework is:

$$\mathcal{L} = \mathcal{L}_{MI} + \beta\mathcal{L}_{ub} + \gamma\mathcal{L}_{ba} + \eta\mathcal{L}_{ta} \tag{21}$$

where $\beta$, $\gamma$ and $\eta$ denote the hyperparameter to balance each component.

### 3.6 Theoretical Analysis

**Theorem 3.1.** *(Fairness Upper Bound)*

*Let classifier $h$ depend only on $\boldsymbol{Z}_c$. Assume $h$ is L-lipschitz in distribution shift. Define*

$$\Delta_{EO} = \sum_{y \in \{0,1\}} |P(h = 1|S = 0, Y = y) - P(h = 1|S = 1, Y = y)|.$$

*If (1) $I(\boldsymbol{Z}_s; \boldsymbol{Z}_c) \leq \epsilon$, and (2) $I(\boldsymbol{Z}_c; Y) \geq \kappa > 0$,*

*then there exist constants $c_1, c_2 > 0$ such that*

$$\Delta_{EO} \leq c_1 \frac{\sqrt{\epsilon}}{\kappa} + c_2 L.$$

**Theorem 3.2.** *(Fair Domain Adaptation Bound)*

*Let $h$ be a classifier on $\boldsymbol{Z}_c$. Then*

$$\epsilon_{ta}(h) \leq \epsilon_{so}(h) + C \cdot disc(p_{so}(\boldsymbol{Z}_c), p_{ta}(\boldsymbol{Z}_c)) + \lambda^* + c_1 I(\boldsymbol{Z}_s; \boldsymbol{Z}_c),$$

*where disc is a discrepancy measure(e.g., $\mathcal{H}\Delta\mathcal{H}$ divergence),$\lambda^*$ is the joint optimal error, and the last term accounts for residual sensitive leakage.*

Table 1: Classification and fairness metrics ($\%\pm\sigma$) on the Bail, Credit, Pokec and syn datasets. Results are reported as mean $\pm$ standard deviation across runs. $\uparrow$ denotes higher-is-better and $\downarrow$ lower-is-better. The best result in each column is **bold**; the second best is underlined.

| Dataset | Metric | GCN | NIFTY | FairVGNN | FairSIN | SFG | SPA | SGDA | FatraGNN | DANCE | COSTA |
|---|---|---|---|---|---|---|---|---|---|---|---|
| Bail-t | ACC↑ | 81.37±1.76 | 81.18±0.81 | 82.75±1.54 | 80.1±0.55 | 75.64±13.07 | 84.01±2.71 | 74.40±1.11 | 81.84±2.24 | 84.43±1.85 | **94.21±0.04** |
| | ROC-AUC↑ | 95.39±1.32 | 91.20±0.39 | 91.15±0.87 | 96.28±0.91 | 94.35±0.73 | 89.34±2.94 | 81.41±0.78 | 91.88±0.34 | 95.96±0.49 | **97.96±0.06** |
| | $\Delta_{DP}\downarrow$ | 7.37±0.38 | 5.54±0.33 | 11.27±5.55 | 8.07±1.61 | 4.01±1.97 | 5.72±1.45 | 11.50±0.47 | 5.83±0.90 | 3.99±1.05 | **3.88±0.03** |
| | $\Delta_{EO}\downarrow$ | 7.73±0.80 | 5.83±0.97 | 10.47±1.77 | **3.75±2.68** | 5.61±2.91 | 4.41±1.92 | 9.68±0.23 | 10.70±0.41 | 4.29±0.69 | 4.98±0.20 |
| | Rank | 5 | 6 | 9 | 3 | 7 | 4 | 10 | 8 | 2 | 1 |
| German-t | ACC↑ | 56.79±3.07 | 56.27±0.79 | 50.51±0.71 | 55.93±3.54 | 54.98±2.11 | 54.03±8.87 | 57.48±2.09 | 56.39±1.83 | **74.29±2.23** | 64.05±0.27 |
| | ROC-AUC↑ | 62.85±1.51 | 65.31±1.03 | 59.36±0.61 | 69.8±1.18 | 63.20±3.61 | 56.04±6.09 | 61.07±1.14 | 65.99±0.25 | 71.63±2.16 | 66.99±0.08 |
| | $\Delta_{DP}\downarrow$ | 15.04±13.18 | 9.43±9.01 | 3.65±5.16 | 2.37±2.28 | 7.70±5.35 | 4.33±3.06 | 4.85±1.48 | 18.27±5.45 | 24.76±5.70 | **1.83±1.07** |
| | $\Delta_{EO}\downarrow$ | 16.54±13.25 | 9.31±9.63 | 2.90±4.10 | 3.36±1.72 | 7.88±5.41 | 5.35±4.63 | 5.10±3.13 | 17.74±7.44 | 21.89±6.42 | **2.27±1.16** |
| | Rank | 9 | 5 | 4 | 2 | 6 | 7 | 3 | 10 | 8 | 1 |
| Pokec-n | ACC↑ | 68.49±0.33 | 68.08±0.72 | 63.90±2.73 | 63.53±5.84 | 53.61±2.65 | 57.66±1.76 | OOM | 65.22±3.26 | 67.65±0.34 | **68.93±0.20** |
| | ROC-AUC↑ | 76.36±0.24 | 72.68±0.46 | 70.33±0.30 | 70.56±1.03 | 64.02±2.80 | 60.58±2.72 | OOM | 72.66±0.39 | 74.44±0.23 | 76.16±0.11 |
| | $\Delta_{DP}\downarrow$ | 2.76±0.42 | 1.79±0.36 | 3.68±1.78 | 3.73±1.74 | 3.31±3.29 | 3.39±1.00 | OOM | 0.93±0.38 | 5.22±0.84 | **0.37±0.30** |
| | $\Delta_{EO}\downarrow$ | 2.01±0.44 | 1.98±0.36 | 2.56±1.43 | 3.9±1.8 | 2.47±2.60 | 2.95±1.85 | OOM | 1.46±0.85 | 5.54±0.94 | **0.71±0.33** |
| | Rank | 2 | 3 | 6 | 7 | 9 | 8 | 10 | 4 | 5 | 1 |
| syn-t | ACC↑ | 82.50±0.01 | 82.65±0.13 | 82.55±0.03 | 62.39±9.03 | 84.10±0.83 | 70.58±0.99 | 78.45±0.15 | 81.46±1.59 | **86.87±0.34** | 79.70±0.02 |
| | ROC-AUC↑ | 90.63±0.00 | 90.78±0.03 | 90.58±0.10 | 73.97±7.87 | 90.78±0.40 | 76.34±1.62 | 87.72±0.05 | 90.77±0.01 | **92.36±0.08** | 87.64±0.05 |
| | $\Delta_{DP}\downarrow$ | 11.51±0.07 | 12.36±0.27 | 11.01±0.86 | 14.61±10.10 | 26.96±1.44 | 5.14±3.18 | 29.35±0.28 | 13.62±0.05 | 25.34±0.67 | **3.36±0.26** |
| | $\Delta_{EO}\downarrow$ | 8.00±0.08 | 9.64±0.52 | 7.44±0.84 | 14.8±10.9 | 30.03±5.88 | 6.72±3.98 | 28.27±1.17 | 3.17±0.03 | 23.13±1.00 | **0.42±0.27** |
| | Rank | 3 | 4 | 2 | 8 | 6 | 6 | 10 | 7 | 5 | 1 |

**Lemma 3.3.** *(Bias Control with Class-wise Thresholds).*

*Let per-class adaptive thresholds $\tau_k = M_k\tau$ with $M_k = max\{m_v^{ta} : arg\, max\psi(\boldsymbol{z}_{v,c}) = k\}$. Define confident set $C = \{v : m_v^{ta} > \tau_k\}$. Then selection bias satisfies*

$$Bias_{sel} = \sum_k |Pr(v \in C|Y = k) - \rho| \leq \sum_k |Pr(m_v^{ta} > \tau_k|Y = k) - \rho|,$$

*for target coverage $\rho$. Adaptive $\tau_k$ balances coverage across classes, reducing bias while training only on confident samples.*

## 4 EXPERIMENTS

**Datasets & Baselines.** We evaluate COSTA on three enhanced real-world graphs and one synthetic benchmark from Qian et al. Qian et al. (2024): 1) *Credit-Cs* is built from the Credit dataset Yeh & Lien (2009). 2) *Pokecs* is constructed from a Slovak social network Dai & Wang (2022), grouping users by province. 3) *Bail-Bs* is derived from the Bail dataset Jordan & Freiburger (2015), where nodes are defendants released on bail. 4) *Synthetic* test fair GNNs when edges carry signal and topology can amplify bias Qian et al. (2024). We compare COSTA with four groups of baselines: (A) Traditional learning methods, including GCN Kipf & Welling (2016). (B) Fairness-Aware GNNs under independent and identically distributed (IID) settings, including NIFTY Agarwal et al. (2021), FairVGNN Wang et al. (2022), FairSIN Yang et al. (2024), and SFG Chen et al.. (C) General Domain Adaptation Methods, including SGDA Qiao et al. (2023) and SPA Xiao et al. (2023). (D) Fairness-Aware GNNs under Out-of-Distribution (OOD) Settings, including FatraGNN Li et al. (2024) and DANCE Wang et al.. More details about the experimental settings are provided in Appendix B.

**Performance Evaluation.** We evaluate node classification performance using two primary metrics: accuracy (ACC) and ROC_AUC. Fairness is assessed using $\Delta_{DP}$ and $\Delta_{EO}$, as defined in Section 2, with lower values indicating better fairness. To comprehensively evaluate classification and fairness, we adopt a composite metric: $c = \text{ACC} + \text{ROC\_AUC} - \Delta_{DP} - \Delta_{EO}$, where higher values indicate better overall performance. The final score for each method is obtained by summing its scores in target domain, and overall rankings are reported accordingly.

### 4.1 PERFORMANCE ANALYSIS

Table 1 report the best average performance of all methods across four datasets. Several key observations can be drawn: (1) When comparing traditional learning methods with Fair GNNs, we observe that fairness-oriented models improve fairness performance at the cost of classification accuracy. Moreover, as Fair GNNs are designed for IID settings, they often struggle under domain

Table 2: Ablation studies on the variants of COSTA.

| Variant | | Pokec-n | | | | syn-t | | |
|---|---|---|---|---|---|---|---|---|
| Metric | ACC↑ | ROC-AUC↑ | $\Delta_{DP}$ ↓ | $\Delta_{EO}$ ↓ | ACC↑ | ROC-AUC↑ | $\Delta_{DP}$ ↓ | $\Delta_{EO}$ ↓ |
| Var1 | **69.79±0.18** | **76.97±0.21** | 0.73±0.26 | **0.26±0.08** | 77.83±4.51 | 85.46±4.91 | 9.01±4.85 | 7.65±6.41 |
| Var2 | 69.14±0.15 | 75.85±0.15 | 0.79±0.38 | 0.93±0.50 | 79.47±0.15 | 87.47±0.05 | **2.60±0.17** | 1.47±0.33 |
| Var3 | 69.07±0.11 | 75.78±0.17 | 0.95±0.08 | 1.17±0.17 | 80.43±0.04 | 88.46±0.04 | 5.85±0.14 | 2.22±0.22 |
| Var4 | 68.66±0.13 | 75.86±0.04 | 1.22±0.04 | 1.01±0.06 | **81.56±0.05** | **89.81±0.02** | 10.30±0.43 | 7.25±0.36 |
| COSTA | 68.93±0.20 | 76.16±0.11 | **0.37±0.30** | 0.71±0.33 | 79.70±0.02 | 87.64±0.05 | 3.36±0.26 | **0.42±0.27** |

Figure 2: Utility and fairness comparison w.r.t. different top-$K$ for retrieval node from source domain and different values for threshold parameter $\tau$ and $\delta$.

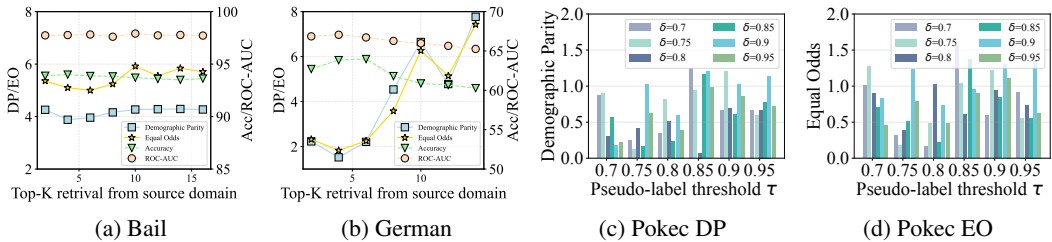

(a) Bail      (b) German      (c) Pokec DP      (d) Pokec EO

adaptation, exhibiting an even poorer trade-off between accuracy and fairness compared to vanilla GCNs. (2) When comparing Fair GNNs with fairness-aware GNNs designed for OOD settings, we observe that OOD methods achieve a better balance between classification and fairness under domain adaptation, owing to their ability to learn fair representations across varying distributions. This highlights the necessity of incorporating domain adaptation modules in fairness-aware GNNs. (3) When comparing fairness-aware GNNs under OOD settings with general domain adaptation methods, we find that the latter often achieve either high performance or strong fairness, but struggle to balance both. This underscores the scarcity of fairness-aware GNN approaches specifically designed for domain adaptation scenarios. (4) COSTA outperforms all other baselines in most cases. Compared to fairness-aware GNNs under IID and OOD settings, as well as general domain adaptation methods, COSTA achieves a better balance between accuracy and fairness in domain adaptation, demonstrating the effectiveness of its domain adaptation module design.

## 4.2 ABLATION STUDY

To assess each module in COSTA and its contribution, we compare COSTA with four ablations: (1) Variant 1: Replaces the MI-based GNN encoder with a vanilla GCN. (2) Variant 2: Removes the bipartite-aware domain alignment module. (3) Variant 3: Removes the sensitive consistency learning module. (4) Variant 4: Removes pseudo-label supervision and uses only ground-truth labels. Table 2 reports results, with key observations as follows: (1) Compared to Variant 1, replacing the MI-based GNN with a vanilla GCN significantly degrades both utility and fairness. The MI objective yields fairness-aware embeddings that enable clean cross-domain top-$k$ retrieval. Without MI-based GNN, representation leaks sensitive information, biases edge selection, and weakens alignment and consistency training. (2) The comparison with Variant 2 demonstrates that explicitly linking same-label source–target nodes and regularizes $\boldsymbol{PP}^T$ shrinks the cross-domain gap. This stabilizes pseudo-labels, reduces biased edge propagation, and yields better alignment thereby improving accuracy and fairness. (3) Compared with Variant 3, removing the sensitive consistency loss worsens fairness and slightly hurts utility. Enforcing sensitive inter-group invariance and target inter-group separation, which curbs sensitive-attribute leakage and keeps embeddings aligned across domains. (4) Compared with Variant 4, removing pseudo-labels harms both utility and fairness. High-confidence pseudo labels are pivotal, because they provide target-side supervision to adapt the decision boundary, define label-consistent alignment pairs and cleaner cross-domain top-$k$ retrieval, and enable the sensitive consistency loss on target nodes. Without them, training is driven only by source labels, cross-domain pairing becomes noisier, and representation drift increases bias.

## 4.3 PARAMETER ANALYSIS

We analyze the impact of three hyperparameter groups in COSTA: (1) The top-$K$ source nodes retrieved based on MI-based GNN; (2) The pseudo label threshold $\tau$ and $\delta$; (3) The loss weights

Figure 3: Fairness comparison w.r.t. different values for loss weights $\gamma$ and $\beta$

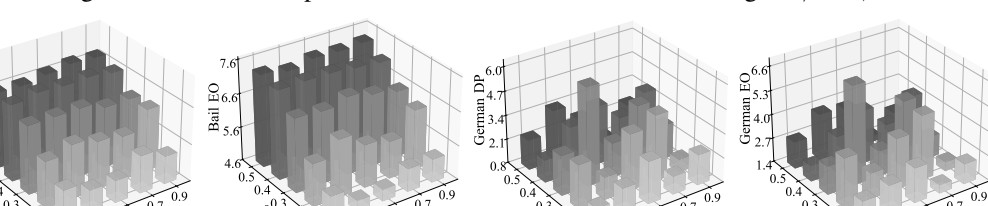

(a) Bail DP      (b) Bail EO      (c) German DP      (d) German EO

Figure 4: Domain adaptation t-SNE: Source-Learned Representations on Pokec-n (target domain)

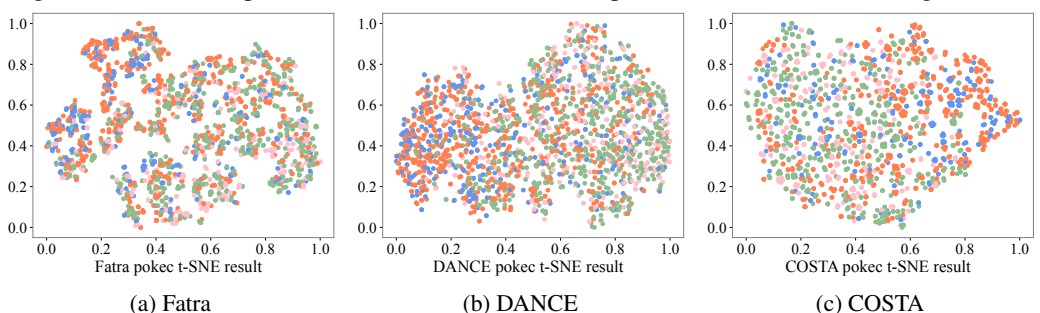

(a) Fatra      (b) DANCE      (c) COSTA

$\beta$ and $\gamma$ in the objective function (Equation 21). Our key findings are as follows: (1) As shown in Figure 5a and 5a, too small top-$K$ under-exploits informative cross-domain links, whereas overly large $K$ injects noisy or demographically imbalanced neighbors. Both degrade utility and fairness on most datasets. Moreover, the optimal $K$ is dataset-specific, varying substantially across Ba, German, Pokec, and syn. (2) As shown in Figure 6c and 6d, moderate–high thresholds improve both utility and fairness by filtering noisy pseudo-labels and sharpening the decision boundary. Effective regions are typically $\delta, \tau \geq 0.75$, though optima are dataset-dependent. Low thresholds admit noise and harm fairness, whereas overly strict ones reduce supervision and can depress utility. (3) As shown in Figure 3, the optimal $(\gamma, \beta)$ settings vary widely across Bail, German, Pokec, and syn, and no single setting dominates on fairness. Some datasets prefer stronger alignment (larger $\gamma$) with weaker fairness regularization (smaller $\beta$), while others exhibit the opposite pattern. This reflects dataset-dependent correlations among labels, sensitive attributes, and topology, necessitating dataset-specific weighting to balance accuracy and fairness.

## 4.4 VISUALIZATION

We visualize the source-trained embeddings on the Pokec-n target graph using t-SNE Van der Maaten & Hinton (2008). Points are colored by the $(Y, S)$ group (target and sensitive labels). As shown in Fig. 4, Fatra exhibits several well-separated islands, indicating stronger alignment of the representation with group identity. DANCE yields more mixing across groups but still shows noticeable cluster boundaries. COSTA produces the most group-mixed manifold with fewer isolated clusters, suggesting reduced sensitive-attribute leakage and improved cross-domain alignment. While t-SNE is qualitative, these patterns are consistent with the observed improvements in fairness metrics (lower DP and EO) without sacrificing utility.

## 5 CONCLUSION

In this paper, we propose a causality-attended representation disentanglement framework with structural alignment framework (COSTA) to address the problem of fairness-aware graph domain adaptation. Moving beyond i.i.d. assumptions, COSTA disentangles task-relevant and sensitive factors via dual encoders with mutual-information methodology, augments target supervision through group-aware pseudo-labeling, and performs fairness-aware bipartite alignment to mitigate spurious correlations across domains. Extensive experiments on three real-world and one synthetic benchmark show consistent gains in both utility and group fairness under distribution shifts. These results position COSTA as a reliable approach to fair graph learning in real-world cross-domain scenarios.

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

## .1 Large Language Models Usage Statement

In the preparation of this research, large language models (LLMs) were employed strictly as a limited-purpose auxiliary tool. The models were used exclusively for language polishing tasks, including grammar checking, sentence structure optimization, and wording refinement to improve the readability and linguistic fluency of portions of the text. The LLMs played no role in any core research activities, including but not limited to: research ideation, theoretical development, experimental design, data analysis, result interpretation, or scientific decision-making. All intellectual contributions to this work originate solely from the human authors. The authors take full responsibility for the entire content of this paper, including text polished by LLMs, and affirm its originality, accuracy, and academic integrity.

## A    Related Work

**Group Fairness in GNNs.** Graph neural networks (GNNs) can inadvertently propagate bias because sensitive attributes are often entangled with graph topology Chen et al. (2024). Fairness in GNNs is typically studied along two axes, namely group fairness and individual fairness, with most work targeting the former by equalizing outcomes across demographic subpopulations. A prominent line for group fairness removes sensitive information from representations via adversarial learning: FairGNN Dai & Wang (2021) trains an auxiliary discriminator and adds a covariance penalty to encourage statistical parity, while FairVGNN Wang et al. (2022) employs a discriminator to hide protected attributes in learned representations. Beyond adversarial debiasing, explicit group-fairness objectives are optimized by jointly minimizing accuracy loss with disparity-based penalties using Lagrangian or multi-objective formulations Chen et al. (2024). For example, NIFTY constructs counterfactual feature views by flipping the sensitive attribute and applies a contrastive loss that pulls original–counterfactual pairs together while pushing apart label-inconsistent pairs Agarwal et al. (2021). Another complementary line debiases inputs structure: For example, FairWalk Rahman et al. (2019) FairWalk reweights random walks to yield demographically balanced contexts EDITS Dong et al. (2022) removes sensitive cues from node features while preserving utility. In summary, these research avenues outline mainstream paradigms—adversarial, constraint-based, contrastive, and structural—for mitigating bias in GNN-based learning.

**Fairness under Distribution Shift.** Distribution shift can significantly degrade fairness when test and training distributions diverge Liu et al. (2021). In response, a growing literature seeks to maintain fairness amidst such shifts. Rezaei et al. (2021); Giguere et al. (2022); Lin et al. (2024). Rezaei et al. (2021) propose a robust-fairness method for covariate shift that adapts models using unlabeled target-domain data. Giguere et al. (2022) focus on demographic shift—changes in subgroup prevalences—and provide high-confidence fairness guarantees when test-time group proportions differ from training. Mandal et al. (2020) adopt a worst-case, distributionally robust approach, modeling the test distribution as a weighted combination of training samples and optimizing fairness under this adversarial shift. An et al. (2022) establish sufficient conditions for fairness transfer and introduce a self-training algorithm with fairness-aware consistency regularization to maintain group fairness across source and target domains. For a more full review for fairness under distribution shifts, please refer to a recent survey Lin et al. (2024). The majority of these approaches ignore relational structure and assume Euclidean data. In order to handle distribution shifts, FatraGNN Li et al. (2024) explicitly addresses graphs by producing additional biased training graphs and minimizing group-wise representation distances between the created and original graphs. In addition, DANCE Wang et al. addresses group imbalance and emphasizes fairness under graph shifts, enhancing fairness under shifting distributions without compromising task performance.

## B    Detailed Experimental Settings

**Datasets.** We evaluate COSTA on three enhanced real-world graphs and one synthetic benchmark from Qian et al. Qian et al. (2024): 1) *Credit-Cs* is built from the Credit dataset Yeh & Lien (2009), in which nodes represent credit card users. The task is binary credit-risk prediction with *age* as the sensitive attribute. We apply modularity-based community detection Newman (2006) to split the graph into *credit-s* as the source domain and *credit-t* as the target domain with distinct distributions. 2) *Pokecs* is constructed from a Slovak social network Dai & Wang (2022), grouping users by province.

The task predicts users' working fields and the sensitive attribute is *region*. It consists of two graphs: *Pokec-z* and *Pokec-n*, where the former is used as the source domain and the latter as the target domain. 3) *Bail-Bs* is derived from the Bail dataset Jordan & Freiburger (2015), where nodes are defendants released on bail. The task is to decide whether a defendant should be granted bail and the sensitive attribute is *race*. As the same with Credit-Cs, community detection yields *Bail-s* as the source domain and *Bail-t* as the target domain. 4) *Synthetic* test fair GNNs when edges carry signal and topology can amplify bias Qian et al. (2024). For each node, $(S, Y)$ are jointly sampled from a categorical distribution with user-specified group proportions. Features concatenate two multivariate Gaussians conditioned on $S$ and $Y$ with tunable means/variances. Each edge type is generated independently via its own Bernoulli probability. We use *Syn-2* as source domain and *Syn-1* as target domain.

**Baselines.** We compare COSTA with four groups of baselines: (A) Traditional learning methods: Fundamental graph representation learning approaches, including GCN Kipf & Welling (2016). (B) Fairness-Aware GNNs under independent and identically distributed (IID) settings: GNNs specifically designed to enhance fairness in IID scenarios, including NIFTY Agarwal et al. (2021), FairVGNN Wang et al. (2022), FairSIN Yang et al. (2024), and SFG Chen et al.. (C) General Domain Adaptation Methods: General approaches aimed at learning robust representations for domain adaptation, including SGDA Qiao et al. (2023) and SPA Xiao et al. (2023). (D) Fairness-Aware GNNs under Out-of-Distribution (OOD) Settings: Graph neural network methods specifically designed to address distribution shifts while preserving fairness between training and test distributions, such as FatraGNN Li et al. (2024) and DANCE Wang et al..

# C  ADDITIONAL RESULTS

## C.1  EXPERIMENTAL SETTING.

During the experiments, we perform hyperparameter tuning via grid search across all dataset groups to ensure a fair and comprehensive evaluation. For COSTA, the embedding dimension is set to $64$. We explore the number of graph encoder layers in the range of $[2, 4]$, dropout rates between $[0, 0.5]$, and learning rates in $[0.002, 0.006]$. To ensure robustness, each method is evaluated over five independent runs with different random seeds, and the mean and variance of each metric are reported.

## C.2  MORE ABLATION STUDY RESULTS

The results of the complete ablation studies are presented in 3 and 4.

Table 3: Ablation studies on the variants of COSTA.

| Variant | Bail-t | | | | German-t | | | |
|---|---|---|---|---|---|---|---|---|
| Metric | ACC↑ | ROC-AUC↑ | $\Delta_{DP}\downarrow$ | $\Delta_{EO}\downarrow$ | ACC↑ | ROC-AUC↑ | $\Delta_{DP}\downarrow$ | $\Delta_{EO}\downarrow$ |
| Var1 | 92.58±0.95 | 97.06±0.26 | **3.42±0.75** | 5.45±0.40 | 63.26±0.08 | **67.34±0.12** | 5.14±0.80 | 9.75±0.26 |
| Var2 | 93.72±0.18 | 97.81±0.10 | 4.16±0.08 | 5.59±0.14 | **64.11±0.41** | 67.04±0.13 | 2.77±1.58 | 3.10±1.65 |
| Var3 | 93.81±0.03 | 97.66±0.08 | 4.09±0.04 | 5.06±0.25 | 63.63±0.23 | 66.92±0.15 | 2.65±1.60 | 3.38±1.71 |
| Var4 | 93.78±0.13 | 97.83±0.08 | 4.12±0.03 | 5.37±0.18 | 63.75±0.62 | 66.99±0.09 | 2.22±1.46 | 2.30±1.16 |
| COSTA | **94.21±0.04** | **97.96±0.06** | 3.88±0.03 | 4.98±0.20 | 64.05±0.27 | 66.99±0.08 | **1.83±1.07** | **2.27±1.16** |

Table 4: Ablation studies on the variants of COSTA.

| Variant | Pokec-n | | | | syn-t | | | |
|---|---|---|---|---|---|---|---|---|
| Metric | ACC↑ | ROC-AUC↑ | $\Delta_{DP}\downarrow$ | $\Delta_{EO}\downarrow$ | ACC↑ | ROC-AUC↑ | $\Delta_{DP}\downarrow$ | $\Delta_{EO}\downarrow$ |
| Var1 | **69.79±0.18** | **76.97±0.21** | 0.73±0.26 | **0.26±0.08** | 77.83±4.51 | 85.46±4.91 | 9.01±4.85 | 7.65±6.41 |
| Var2 | 69.14±0.15 | 75.85±0.15 | 0.79±0.38 | 0.93±0.50 | 79.47±0.15 | 87.47±0.05 | **2.60±0.17** | 1.47±0.33 |
| Var3 | 69.07±0.11 | 75.78±0.17 | 0.95±0.08 | 1.17±0.17 | 80.43±0.04 | 88.46±0.04 | 5.85±0.14 | 2.22±0.22 |
| Var4 | 68.66±0.13 | 75.86±0.04 | 1.22±0.04 | 1.01±0.06 | **81.56±0.05** | **89.81±0.02** | 10.30±0.43 | 7.25±0.36 |
| COSTA | 68.93±0.20 | 76.16±0.11 | **0.37±0.30** | 0.71±0.33 | 79.70±0.02 | 87.64±0.05 | 3.36±0.26 | **0.42±0.27** |

### C.3 MORE PARAMETER ANALYSIS RESULTS

The results of the complete parameter analysis are presented in 6 and 7.

Figure 5: Utility and fairness comparison w.r.t. different top-$K$ for retrieval node from source domain.

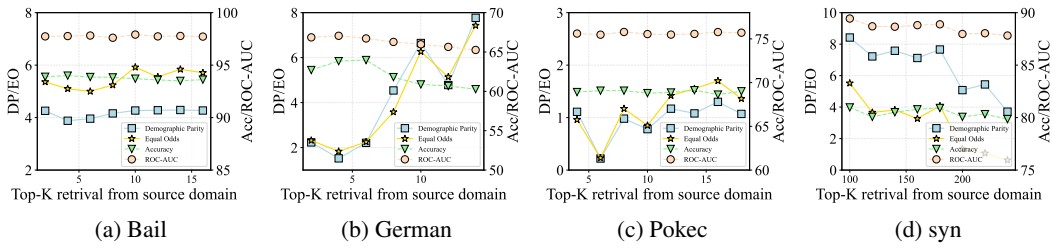

(a) Bail      (b) German      (c) Pokec      (d) syn

Figure 6: Fairness comparison w.r.t. different values for threshold parameter $\tau$ and $\delta$

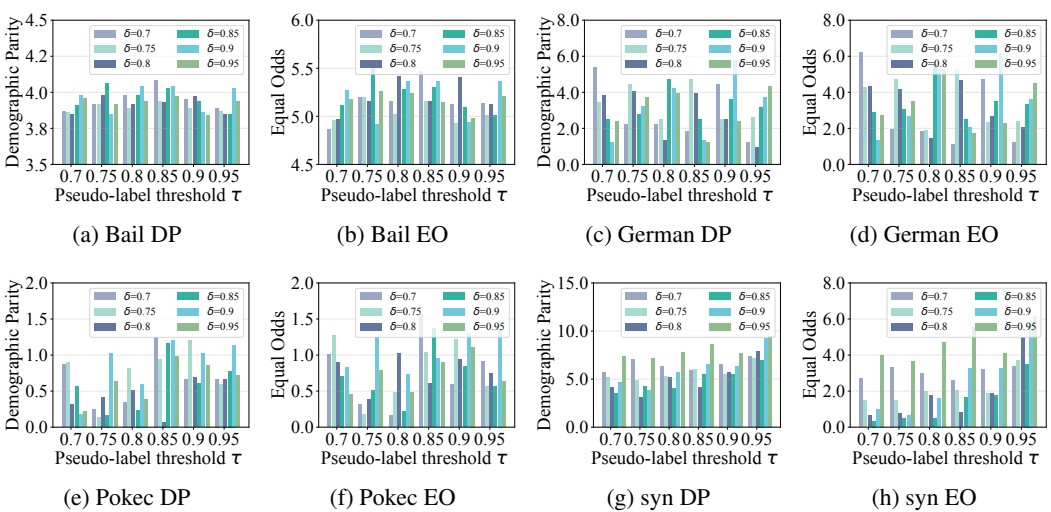

(a) Bail DP      (b) Bail EO      (c) German DP      (d) German EO

(e) Pokec DP      (f) Pokec EO      (g) syn DP      (h) syn EO

Figure 7: Fairness comparison w.r.t. different values for loss weights $\gamma$ and $\beta$

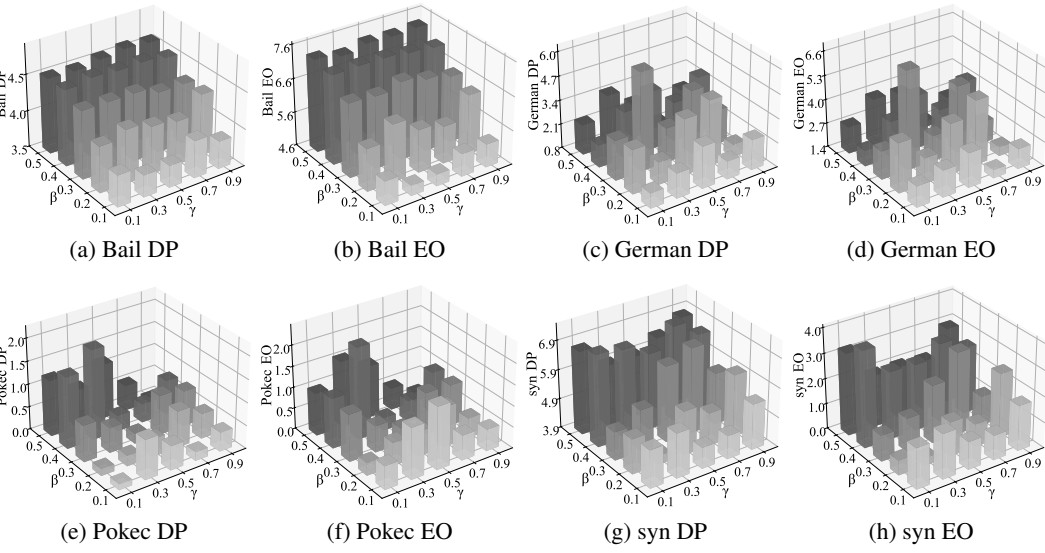

(a) Bail DP      (b) Bail EO      (c) German DP      (d) German EO

(e) Pokec DP      (f) Pokec EO      (g) syn DP      (h) syn EO

# D    PROOF OF THEORETICAL RESULTS

**Theorem D.1.** *(CLUB Upper Bound for $I(\mathbf{Z}_s; \mathbf{Z}_c)$).*

*For any conditional density estimator $q_\theta(\mathbf{z}_s|\mathbf{z}_c)$,define*

$$CLUB_\theta := \mathbb{E}_{p(\mathbf{z}_s|\mathbf{z}_c)}[logq_\theta(\mathbf{z}_s|\mathbf{z}_c)] - \mathbb{E}_{p(\mathbf{z}_c)p(\mathbf{z}_s)}[logq_\theta(\mathbf{z}_s|\mathbf{z}_c)].$$

*Then,*

$$I(\mathbf{Z}_s; \mathbf{Z}_c) \leq CLUB_\theta + \mathbb{E}_{p(\mathbf{z}_s,\mathbf{z}_c)}log\frac{p(\mathbf{z}_s|\mathbf{z}_c)}{q_\theta(\mathbf{z}_s|\mathbf{z}_c)}.$$

*Hence,$CLUB_\theta$ is an upper bound on $I(\mathbf{Z}_s; \mathbf{Z}_c)$; the bound tightens as $q_\theta \to p(\cdot|\cdot)$.*

*Proof.* By definition,

$$I(\mathbf{Z}_s; \mathbf{Z}_c) = \mathbb{E}_{p(\mathbf{z}_s,\mathbf{z}_c)}[\log\frac{p(\mathbf{z}_s|\mathbf{z}_c)}{p(\mathbf{z}_s)}].$$

For any variational distribution $q_\theta$, we add and subtract $logq_\theta(\mathbf{z}_s, \mathbf{z}_c)$:

$$I(\mathbf{Z}_s; \mathbf{Z}_c) = \left(\mathbb{E}_{p(\mathbf{z}_s,\mathbf{z}_c)} \log q_\theta(\mathbf{z}_s \mid \mathbf{z}_c) - \mathbb{E}_{p(\mathbf{z}_c)p(\mathbf{z}_s)} \log q_\theta(\mathbf{z}_s \mid \mathbf{z}_c)\right)$$

$$+ \mathbb{E}_{p(\mathbf{z}_s,\mathbf{z}_c)}\left[\log \frac{q_\theta(\mathbf{z}_s \mid \mathbf{z}_c)}{p(\mathbf{z}_s \mid \mathbf{z}_c)}\right]$$

$$= \text{CLUB}_\theta + \mathbb{E}_{p(\mathbf{z}_c)} \text{KL}\big(p(\cdot \mid \mathbf{z}_c) \,\|\, q_\theta(\cdot \mid \mathbf{z}_c)\big). \tag{22}$$

Since the KL term is always non-negative, the inequality follows. Equality holds when $q_\theta = p$.  $\square$

**Theorem D.2.** *(Conditional InfoNCE as a Lower Bound on $I(\mathbf{Z}_C; D|\mathbf{Z}_s)$)*

*Let $f(\boldsymbol{\mu}, \mathbf{z}_c, \mathbf{z}_s)$be a bounded scoring function. For batch size $M$, define*

$$\mathcal{L}_{cNCE} := -\mathbb{E}[log\frac{expf(\boldsymbol{\mu}_v, \mathbf{z}_{v,c}, \mathbf{z}_{v,s})}{\frac{1}{M}\sum_{u=1}^{M} expf(\boldsymbol{\mu}_u, \mathbf{z}_{v,c}, \mathbf{z}_{v,s})}].$$

*Then*

$$I(\mu; \mathbf{Z}_c|\mathbf{Z}_s) \geq -\mathcal{L}_{cNCE}.$$

*If the anchor $\boldsymbol{\mu}$ encodes domain-related information, there exists $\kappa > 0$ such that $I(\mathbf{Z}_c; D|\mathbf{Z}_x) \geq -\kappa\mathcal{L}_{cNCE}$.*

*Proof.* Given $C = c$, build a contrastive set $\mathcal{S} = \{(\boldsymbol{\mu}_u, \mathbf{z}_{u,c})\}_{u=1}^{M}$ containing one positive from $p_+(\cdot \mid c)$ and $M - 1$ negatives from $p_-(\cdot \mid c)$. Let $K \in \{1, \ldots, M\}$ be the (uniform) index of the positive pair. Define the softmax classifier

$$q_f(k \mid \mathcal{S}, c) = \frac{\exp f(\boldsymbol{\mu}_k, \mathbf{z}_{k,c}, c)}{\sum_{i=1}^{M} \exp f(\boldsymbol{\mu}_i, \mathbf{z}_{i,c}, c)}.$$

The usual sum-denominator InfoNCE cross-entropy is $\mathcal{L}_{\text{sum}} = \mathbb{E}[-\log q_f(K \mid \mathcal{S}, C)]$, and our avg loss satisfies the identity $\mathcal{L}_{\text{cNCE}} = \mathcal{L}_{\text{sum}} - \log M$.

At the Bayes score $f^\star = \log \frac{p_+}{p_-}$, $q_{f^\star} = p(K \mid \mathcal{S}, C)$, hence

$$\min_f \mathcal{L}_{\text{sum}} = \mathbb{E}[-\log p(K \mid \mathcal{S}, C)] = \log M - I(K; \mathcal{S} \mid C),$$

since $H(K \mid C) = \log M$ and $I(K; \mathcal{S} \mid C) = H(K \mid C) - H(K \mid \mathcal{S}, C)$. For any $f$ (by Gibbs' inequality), $\mathcal{L}_{\text{sum}} \geq \log M - I(K; \mathcal{S} \mid C)$. By data processing in the sampling scheme (the only dependence of $K$ on $\mathcal{S}$ flows through the joint information in $(\boldsymbol{\mu}, \mathbf{Z}_c)$), $I(K; \mathcal{S} \mid C) \leq I(\boldsymbol{\mu}; \mathbf{Z}_c \mid C)$. Thus

$$\mathcal{L}_{\text{sum}} \geq \log M - I(\boldsymbol{\mu}; \mathbf{Z}_c \mid C) \iff I(\boldsymbol{\mu}; \mathbf{Z}_c \mid C) \geq \log M - \mathcal{L}_{\text{sum}}.$$

Using $\mathcal{L}_{\text{cNCE}} = \mathcal{L}_{\text{sum}} - \log M$ gives the stated avg-form bound: $I(\boldsymbol{\mu}; \mathbf{Z}_c \mid C) \geq -\mathcal{L}_{\text{cNCE}}$.  $\square$

**Theorem D.3.** *(Fairness Upper Bound)*

*Let classifier $h$ depend only on $\mathbf{Z}_c$. Assume $h$ is $L$-lipschitz in distribution shift. Define*

$$\Delta_{EO} = \sum_{y \in \{0,1\}} |P(h = 1|S = 0, Y = y) - P(h = 1|S = 1, Y = y)|.$$

*If (1) $I(\mathbf{Z}_s; \mathbf{Z}_c) \leq \epsilon$, and (2) $I(\mathbf{Z}_c; Y) \geq \kappa > 0$,*

*then there exist constants $c_1, c_2 > 0$ such that*

$$\Delta_{EO} \leq c_1 \frac{\sqrt{\epsilon}}{\kappa} + c_2 L.$$

*Proof.* Fix $y \in \{0, 1\}$ and let

$$P_y := \mathcal{L}(\mathbf{Z}_c|S = 0, Y = y), \quad Q_y = \mathcal{L}(\mathbf{Z}_c|S = 1, Y = y).$$

Let $A_h := \{\mathbf{z}_c : h(\mathbf{z}_c) = 1\}$. Then

$$|\Pr(h = 1|S = 0, Y = y) - \Pr(h = 1|S = 1, Y = y)| = |P_y(A_h) - Q_y(A_h)| \leq \mathrm{TV}(Q_y, Q_y),$$

hence

$$\Delta_{\mathrm{EO}} \leq \sum_{y \in \{0,1\}} \mathrm{TV}(P_y, Q_y).$$

For binary $S$, standard inequalities(Pinsker's inequality) relating TV and KL plus the identity$I(\mathbf{Z}_s; S|Y = y) = \sum_{s \in \{0,1\}} \Pr(s|y)\mathrm{KL}(\mathcal{L}(\mathbf{Z}_c|s, y)\|\mathcal{L}(\mathbf{Z}_c|y))$,give(absorbing the class-imbalance factor into a constant $C_y$)

$$\mathrm{TV}(P_y, Q_y) \leq C_y \sqrt{I(\mathbf{Z}_c; S|Y = y)}.$$

Summing over $y$ and using $\sqrt{a} + \sqrt{b} \leq \sqrt{2(a + b)}$yields

$$\sum_y \mathrm{TV}(P_y, Q_y) \leq C\sqrt{I(\mathbf{Z}_c; S|Y)}$$

for a constant $C > 0$ that depends only on label/group proportions.

Since $\mathbf{Z}_s$ is a proxy for $S$, by data processing and a near-sufficiency argument there exists a small $\delta \geq 0$ such that

$$I(\mathbf{Z}_c; S|Y) \leq I(\mathbf{Z}_c; \mathbf{Z}_s|Y) + \delta \leq I(\mathbf{Z}_c; \mathbf{Z}_s) + \delta \leq \epsilon + \delta.$$

So

$$\Delta_{\mathrm{EO}} \leq C\sqrt{\epsilon + \delta}.$$

Assumption $I(\mathbf{Z}_c; Y) \geq \kappa$ says $\mathbf{Z}_c$ carries at least $\kappa$ nats of task signal, Standard stability/margin arguments (e.g., calibrated link or strong-convexity of the surrogate loss) imply that the contribution of spurious variation in $\mathbf{Z}_c$ to the decision probability is attenuated by a factor proportional to $\frac{1}{\kappa}$.Thus there is a constant $c_1 > 0$ such that

$$\Delta_{\mathrm{EO}} \leq c_1 \frac{\sqrt{\epsilon + \delta}}{\kappa} + c_2 L,$$

where the additive $c_2 L$term accounts for the $L$-Lipschitz sensitivity of $h$ under residual distributional shift not captured by the MI control(a standard device to prevent amplification when mapping input distributions to predictions).

Finally, absorb $\delta$ into $\epsilon$ and rename constants to get

$$\Delta_{\mathrm{EO}} \leq c_1 \frac{\sqrt{\epsilon}}{\kappa} + c_2 L$$

as claimed. $\square$

**Lemma D.4.** *(Spectral Consistency)*

*Let bipartite adjacency $B$ connect source-target pairs with identical task labels but different sensitive labels. Define normalizes Laplacian $L$ and prediction matrix $P$.Then*

$$L_{ba} = ||(I - L) - PP^T||_F^2 = -2 \sum_{u,v} \frac{B_{uv}}{\sqrt{d_u d_v}} \boldsymbol{p}_u^T \boldsymbol{p}_v + \sum_{u,v} B_{uv} (\boldsymbol{p}_u^T \boldsymbol{p}_v)^2 + const.$$

*Proof.* Expand the Frobenius norm:

$$L_{\mathrm{ba}} = \|\mathbf{A}\|_F^2 + \|\mathbf{PP}^\top\|_F^2 - 2 \operatorname{tr}(\mathbf{A}^\top \mathbf{PP}^\top)$$
$$= \|\mathbf{A}\|_F^2 + \|\mathbf{PP}^\top\|_F^2 - 2 \operatorname{tr}(\mathbf{P}^\top \mathbf{AP}).$$

(i) For the trace term:

$$\operatorname{tr}(\mathbf{P}^\top \mathbf{AP}) = \sum_{u,v} A_{uv} \mathbf{p}_u^\top \mathbf{p}_v = \sum_{u,v} \frac{B_{uv}}{\sqrt{d_u d_v}} \mathbf{p}_u^\top \mathbf{p}_v.$$

(ii) For the squared term:

$$\|\mathbf{PP}^\top\|_F^2 = \sum_{u,v} \left((\mathbf{PP}^\top)_{uv}\right)^2 = \sum_{u,v} (\mathbf{p}_u^\top \mathbf{p}_v)^2.$$

If we only accumulate over edges $(u, v)$ with $B_{uv} = 1$ (as in Eq. (16) of the main text).

(iii) Under the constraint $\mathbf{P}^\top \mathbf{P} = \mathbf{I}_K$, we have $\|\mathbf{PP}^\top\|_F^2 = K$ is constant. Therefore minimizing $L_{\mathrm{ba}}$ is equivalent to maximizing $\operatorname{tr}(\mathbf{P}^\top \mathbf{AP})$, which by the Rayleigh–Ritz theorem is achieved by choosing $\mathbf{P}$'s columns as the top $K$ eigenvectors of $\mathbf{A}$. Equivalently, these correspond to the bottom $K$ eigenvectors of $\mathbf{L}$, consistent with standard spectral clustering. $\square$

**Lemma D.5.** *(Effect of Group-wise Contrastive Losses).*

*Partition neighbors into IG/SG/TG/TSG sets. Define losses $L_{scl}$ and $L_{dis}$ as in the main text. Then:*

*(1)Minimizing $L_{scl}$ increase a lower bound of $I(\boldsymbol{Z}_c; Y)$ by treating IG and SG as positives, TG as negatives.*

*(2)Minimizing $L_{dis}$ reduces effective $I(\boldsymbol{Z}_c; \boldsymbol{Z}_s)$ by penalizing alignment between task and sensitive embeddings.*

*Proof.* Let $Z_c$ denote task representations and $Z_s$ denote sensitive representations. For each anchor $i$ with label $Y_i$, partition neighbors into IG/SG/TG/TSG; in the contrastive loss $L_{\mathrm{scl}}$ we treat IG∪SG as positives and TG as negatives (TSG may be folded into TG as hard negatives). With a similarity score $\operatorname{sim}(\cdot, \cdot)$, $L_{\mathrm{scl}}$ follows the InfoNCE form where one positive $j^+$ is sampled from an approximation of the class-conditional $p(z^c \mid Y_i)$ and $N-1$ negatives $\{j^-\}$ are sampled from the marginal $p(z^c)$. In this $N$-way discrimination, the Bayes-optimal score is a log-likelihood ratio $\log p(z^c \mid Y) - \log p(z^c)$, yielding the standard mutual-information lower bound

$$I(Z_c; Y) \geq \log N - L_{\mathrm{scl}} - \varepsilon,$$

where $\varepsilon$ collects finite-sample and sampling-mismatch errors. Thus minimizing $L_{\mathrm{scl}}$ increases a lower bound on $I(Z_c; Y)$ under the IG/SG-positive and TG-negative construction.

To reduce dependence between $Z_c$ and $Z_s$, the decorrelation loss $L_{\mathrm{dis}}$ penalizes their alignment. Assuming batchwise standardization (so $\operatorname{Cov}(Z_c) = \operatorname{Cov}(Z_s) = I$) and writing the cross-covariance as $\Sigma_{cs} = \mathbb{E}[Z_c Z_s^\top]$, a concrete choice is

$$L_{\mathrm{dis}} \propto \|\Sigma_{cs}\|_F^2 \quad \text{or} \quad \mathbb{E}\left[((z_i^c)^\top z_j^s)^2\right] \quad \text{over IG/SG pairs,}$$

both shrinking the singular values $\{\sigma_r\}$ of $\Sigma_{cs}$ toward zero. Under a Gaussian (whitened CCA) approximation,

$$I(Z_c; Z_s) = -\tfrac{1}{2} \sum_r \log\left(1 - \sigma_r^2\right),$$

which is monotone in the canonical correlations $\{\sigma_r\}$. Hence minimizing $L_{\mathrm{dis}}$ decreases a tight surrogate of $I(Z_c; Z_s)$ and suppresses residual sensitive leakage in $Z_c$.

*Intuition.* $L_{\mathrm{scl}}$ pulls together same-class neighbors (IG/SG) and pushes away task-mismatched ones (TG), thereby maximizing task information in $Z_c$; $L_{\mathrm{dis}}$ orthogonalizes the task and sensitive subspaces, cutting shared variation and lowering effective dependence between $Z_c$ and $Z_s$. $\quad\square$

**Theorem D.6.** *(Fair Domain Adaptation Bound)*

*Let $h$ be a classifier on $\boldsymbol{Z}_c$. Then*

$$\epsilon_{ta}(h) \leq \epsilon_{so}(h) + C \cdot disc(p_{so}(\boldsymbol{Z}_c), p_{ta}(\boldsymbol{Z}_c)) + \lambda^* + c_1 I(\boldsymbol{Z}_s; \boldsymbol{Z}_c),$$

*where disc is a discrepancy measure(e.g., $\mathcal{H}\Delta\mathcal{H}$ divergence),$\lambda^*$ is the joint optimal error, and the last term accounts for residual sensitive leakage.*

*Proof.* By the domain adaptation theorem of Ben-David et al., for any classifier $h$ on the representation space $Z_c$,

$$\epsilon_{ta}(h) \;\leq\; \epsilon_{so}(h) \;+\; C \cdot \mathrm{disc}\big(p_{so}(\boldsymbol{Z}_c),\, p_{ta}(\boldsymbol{Z}_c)\big) \;+\; \lambda^*.$$

Since $p(\boldsymbol{Z}_c) = \sum_s p(\boldsymbol{Z}_c \mid S = s)\, p(S = s)$, the discrepancy term depends on shifts in the sensitive prior $p(S)$. By Pinsker's inequality,

$$\mathrm{TV}\big(p(\boldsymbol{Z}_c \mid s),\, p(\boldsymbol{Z}_c)\big) \;\leq\; \sqrt{\tfrac{1}{2}\,\mathrm{KL}\big(p(\boldsymbol{Z}_c \mid s) \,\|\, p(\boldsymbol{Z}_c)\big)}.$$

Averaging over $s$ yields

$$\mathbb{E}_s\,\mathrm{TV}\big(p(\boldsymbol{Z}_c \mid s),\, p(\boldsymbol{Z}_c)\big) \;\leq\; \sqrt{\tfrac{1}{2}\,I(\boldsymbol{Z}_s; \boldsymbol{Z}_c)}.$$

Hence the residual sensitivity leakage contributes an additional error bounded by the mutual information term, and thus

$$\epsilon_{ta}(h) \;\leq\; \epsilon_{so}(h) \;+\; C \cdot \mathrm{disc}\big(p_{so}(\boldsymbol{Z}_c), p_{ta}(\boldsymbol{Z}_c)\big) \;+\; \lambda^* \;+\; c_1 I(\boldsymbol{Z}_s; \boldsymbol{Z}_c).$$

$\square$

**Lemma D.7.** *(Bias Control with Class-wise Thresholds).*

*Let per-class adaptive thresholds $\tau_k = M_k\tau$ with $M_k = max\{m_v^{ta} : arg\,max\psi(\boldsymbol{z}_{v,c}) = k\}$.Define confident set $C = \{v : m_v^{ta} > \tau_k\}$.Then selection bias satisfies*

$$Bias_{sel} = \sum_k |Pr(v \in C | Y = k) - \rho| \leq \sum_k |Pr(m_v^{ta} > \tau_k | Y = k) - \rho|,$$

*for target coverage $\rho$. Adaptive $\tau_k$ balances coverage across classes, reducing bias while training only on confident samples.*

*Proof.* Let the target-domain maximum posterior confidence be $m_v^{ta} = \max_c \psi(z_v, c)$. For each class $k$, define the per-class scale $M_k = \max\{ m_v^{ta} : \arg\max_c \psi(z_v, c) = k \}$ and the class-wise threshold $\tau_k = M_k \tau$ with $\tau \in (0, 1)$ as a global baseline. Define the confident set $C = \{ v : m_v^{ta} > \tau_k \}$, fix a target coverage $\rho \in (0, 1)$, and write the selection bias as

$$\mathrm{Bias}_{\mathrm{sel}}(\tau) = \sum_k \big| \Pr(v \in C \mid Y = k) - \rho \big|.$$

By the definition of class-wise thresholding, under $Y = k$ the events $\{v \in C\}$ and $\{m_v^{ta} > \tau_k\}$ coincide. Hence

$$\Pr(v \in C \mid Y = k) \;=\; \Pr(m_v^{ta} > \tau_k \mid Y = k), \qquad \mathrm{Bias}_{\mathrm{sel}}(\tau) \;=\; \sum_k \big| \Pr(m_v^{ta} > \tau_k \mid Y = k) - \rho \big|.$$

Conditioned on $Y = k$, introduce the normalized score $R_k = m_v^{ta}/M_k$ and denote its CDF by $F_k(t) = \Pr(R_k \leq t \mid Y = k)$. Then, for any $\tau \in (0, 1)$,

$$\Pr(m_v^{ta} > \tau_k \mid Y = k) \;=\; \Pr(R_k > \tau \mid Y = k) \;=\; 1 - F_k(\tau).$$

Let $F$ be a reference distribution satisfying $F(\tau) = 1 - \rho$ (e.g., the empirical mixture of $\{R_k\}$ used to set $\tau$). It follows that

$$\text{Bias}_{\text{sel}}(\tau) \;=\; \sum_k \left|1 - F_k(\tau) - \rho\right| \;=\; \sum_k \left|F(\tau) - F_k(\tau)\right|.$$

Define the (class-wise) Kolmogorov distances $\delta_k = \sup_t |F_k(t) - F(t)|$. Then

$$\text{Bias}_{\text{sel}}(\tau) \;\leq\; \sum_k \delta_k.$$

$\square$

