# OpenReview forum: "GREAT: GROUP-ACQUIRED BIPARTITE ALIGNMENT FOR GRAPH FAIRNESS ADAPTATION"
_ICLR.cc/2026/Conference — Submitted to ICLR 2026_

### Official Review · Reviewer_mKRG · 2025-10-24

**Soundness:** 3
**Presentation:** 2
**Contribution:** 2
**Rating:** 4
**Confidence:** 4

**Summary:**

The paper studies graph fairness adaptation: transferring task and fairness knowledge from a labeled source graph to a completely unlabeled target graph. The proposed method COSTA uses (i) a dual-encoder GNN with mutual-information objectives to disentangle causal/task features from sensitive features, (ii) group-attended pseudo-labeling that partitions target nodes into intra-/inter-group sets to
promote group fairness, and (iii) a fairness-aware bipartite alignment that links source-target nodes with the same task label $y$ but
different sensitive label $s$, followed by clustering on the induced bipartite graph. The paper further provides fairness and adaptation bounds and reports improvements over multiple baselines across four datasets, with ablations supporting each module.

**Strengths:**

+ The motivation is clear. The paper formally defines graph fairness adaptation with unlabeled target graphs and reviews group-fairness metrics $\Delta\mathrm{DP}$ and $\Delta\mathrm{EO}$ with precise definitions.
+ The model design is sound. The bipartite construction connects same-$y$, different-$s$ pairs and aligns them via a Laplacian clustering
  objective. Class-wise thresholds $\tau_k$ and the confident set $\mathcal{C}$ are defined, with a lemma on selection bias.
+ The experiments are comprehensive. Results on four datasets with multiple fairness baselines and four ablations demonstrate the method's effectiveness.

**Weaknesses:**

- The manuscript, title block, and contributions alternately use COSTA while the submission title mentions GREAT. This mismatch is confusing and should be reconciled consistently across the paper.
- Theorems 3.1-3.2 hinge on the Lipschitzness of $h$, non-trivial bounds on $I(Z_s;Z_c)$ and $I(Z_c;Y)$, and a discrepancy measure, but
  proofs/constant dependencies are not detailed. Please clarify which training losses directly control these quantities and how to estimate them empirically.
- The main evaluation reports group fairness ($\Delta\mathrm{DP}/\Delta\mathrm{EO}$) as mean$\pm$std, but significance tests or confidence intervals are not provided, and individual fairness is not evaluated.
- The problem assumes binary labels and a single binary sensitive attribute. How about the generalization to multi-class/multi-attribute settings?
- Error amplification may occur from mis-labeled $s$ or $y$ pseudo-labels, which can corrupt bipartite edges.

**Questions:**

Please refer to the above weaknesses.

---

### Official Review · Reviewer_TZNh · 2025-10-31

**Soundness:** 2
**Presentation:** 3
**Contribution:** 2
**Rating:** 2
**Confidence:** 5

**Summary:**

This paper investigates the problem of graph adaptation while considering fairness constraints. To address this challenge, the authors propose COSTA (Causality-attended Representation Disentanglement with Structural Alignment), a method that builds a causal graph to facilitate the disentanglement of sensitive information from node representations. This process is specifically designed to enhance model fairness. The effectiveness of the proposed method is validated through comprehensive experiments on multiple datasets, which demonstrate its strong performance in terms of both utility and fairness.

**Strengths:**

S1: The paper is well-structured, and the proposed method demonstrates strong performance in both utility and fairness.

S2: The paper addresses a meaningful research problem that has been under-explored in prior work.

**Weaknesses:**

W1: Insufficient Description of Graph Adaptation Setting: The paper operates at the intersection of graph adaptation and fairness, necessitating a clear introduction to both fields. However, the background provided on graph adaptation in Section 2 is insufficient for readers not already expert in this area, making it difficult to fully grasp the problem setting. Furthermore, the paper should clarify the distinction between its proposed approach and existing work on fairness-aware GNNs under Out-of-Distribution (OOD) settings. This would help to better contextualize the paper's specific contributions.

W2: Inconsistent Source-Target Splitting Strategy: The paper employs inconsistent strategies for splitting datasets into source and target domains, which complicates the interpretation of the results. For example, the Credit-Cs dataset is split using modularity-based community detection, while the Pokec dataset is split based on geographic (province) information. This inconsistency leads to an ambiguous definition of what constitutes a "domain shift". Moreover, if the source and target domains are not sufficiently distinct, making source-domain sensitive attributes available during training could be tantamount to data leakage for the target domain. A more principled and unified approach to domain splitting would significantly strengthen the paper's experimental validity.

W3: High Number and Sensitivity of Hyperparameters: The proposed method introduces a significant number of hyperparameters, including the number of top-K source nodes retrieved, the pseudo-label threshold, and the various loss weights in Eq. 21. This reliance on numerous parameters could necessitate extensive tuning for new datasets, potentially hindering the method's practical applicability. Furthermore, the model appears highly sensitive to these choices. As shown in Figure 2b, fairness performance varies significantly with the value of K, and similar sensitivity is observable for other parameters in Figures 3(c) and 3(d), raising concerns about the model's robustness.

W4: Lack of Complexity Analysis: Given the apparent complexity of the proposed method, the paper would be significantly strengthened by a formal analysis of its time and space complexity. This analysis is crucial, as the method's practical value would be questionable if its training time significantly exceeds that of simply training separate fairness-aware models on the source and target domains. Therefore, it is necessary to provide both a theoretical complexity analysis and an empirical comparison of training times against relevant baselines.

W5: Insufficient Literature Review: The Related Work section omits a discussion of relevant studies focused on graph fairness in settings where demographic information is unavailable. While the paper's setting (fairness in the target domain without access to demographic data) may be distinct, it is crucial to review and differentiate from these closely related works to properly situate the paper's contribution. Specifically, studies such as the following should be considered and discussed:

[1] Zhu Y, Li J, Chen L, et al. The devil is in the data: Learning fair graph neural networks via partial knowledge distillation[C]//Proceedings of the 17th ACM International Conference on Web Search and Data Mining. 2024: 1012-1021.

[2] Wang X, Gu T, Bao X, et al. Towards Fair Graph Neural Networks via Graph Counterfactual without Sensitive Attributes[C]//2025 IEEE 41st International Conference on Data Engineering (ICDE). IEEE, 2025: 265-277.

[3] Wang Z, Hoang N, Zhang X, et al. Towards fair graph learning without demographic information[C]//The 28th International Conference on Artificial Intelligence and Statistics. 2025, 258: 2107-2115.

[4] Wang Z, Liu F, Pan S, et al. fairGNN-WOD: Fair Graph Learning Without Demographics[J].

W6: Flawed Evaluation Metric: The composite evaluation metric ($c= ACC + ROC AUC − \Delta_{DP}-\Delta_{EO}$) is methodologically questionable. Linearly combining metrics with different scales and interpretations (e.g., accuracy versus fairness gaps) is an arbitrary approach that lacks clear justification. This direct aggregation can obscure the crucial trade-offs between utility and fairness, potentially leading to misleading conclusions about a model's overall performance. A more standard approach, such as presenting the utility and fairness metrics separately (e.g., in a Pareto front analysis), would provide a more transparent and meaningful evaluation.


Minor points:

The provided link to the source code repository is currently inaccessible (it results in a "file not found" error). Please ensure that the repository is made public and the link is correct.

**Questions:**

See above.

---

### Official Review · Reviewer_Q6Ee · 2025-11-09

**Soundness:** 3
**Presentation:** 3
**Contribution:** 3
**Rating:** 2
**Confidence:** 3

**Summary:**

This paper addresses the critical limitation in existing fairness-aware graph adaptation methods: the unrealistic assumption that sensitive attributes (e.g., gender, race) are observable on the unlabeled target graph. To overcome this, the authors propose COSTA (Causality-attended Representation Disentanglement with Structural Alignment), a novel framework that enables fair knowledge transfer without requiring sensitive attribute labels on the target graph. COSTA constructs a causal graph to model the underlying graph generation mechanism, then employs two specialized encoders: a sensitive encoder to extract sensitive attribute representations and a causal encoder for invariant causal features. The method ensures representation disentanglement by minimizing mutual information between causal and sensitive representations under conditional distributions. Leveraging unlabeled data, it generates pseudo-labels for both target and sensitive attributes, while a fairness-aware bipartite graph mitigates domain shift by guiding unbiased representation alignment. Extensive experiments on standard graph fairness benchmarks demonstrate that COSTA significantly outperforms state-of-the-art baselines in fairness and adaptation performance, validating its practicality for real-world scenarios where sensitive attributes are typically unobserved.

**Strengths:**

1.Addresses a critical limitation of prior work: Unlike existing fairness-aware graph adaptation methods that require known sensitive attributes on the target graph, COSTA operates without this assumption, making it more practical for real-world scenarios where sensitive attributes are often unobservable.

2.Novel causal mechanism: Introduces a causal graph to model the underlying graph generation process, enabling causal representation disentanglement (separating causal features from sensitive attributes) via mutual information minimization.

3.Strong empirical validation: Demonstrates superior performance over baselines on benchmark datasets, confirming effectiveness in fairness-aware adaptation.

**Weaknesses:**

1. Overreliance on Pseudo-Label Quality. The method generates pseudo-labels for both target labels and sensitive attributes to leverage unlabeled data. However, the paper provides no analysis of how pseudo-label errors propagate (e.g., if sensitive attribute pseudo-labels are inaccurate, fairness could worsen). This undermines the core claim of "fairness without known sensitive attributes."

2. Unjustified Causal Graph Assumptions. COSTA builds a "causal graph" to model graph generation mechanisms, but the paper fails to justify the causal structure (e.g., directionality of edges, presence of unobserved confounders). Without evidence that the assumed causal structure matches real-world graph data, the disentanglement mechanism lacks theoretical grounding.

3. Narrow Fairness Metric Evaluation. The paper only reports two fairness metrics (Demographic Parity, Equal Odds) without testing broader fairness criteria (e.g., Equal Opportunity, Predictive Equality). This risks overlooking trade-offs where fairness improves on one metric but degrades on another—common in real-world deployments.

**Questions:**

1. How does COSTA mitigate performance degradation when pseudo-label error rates for sensitive attributes exceed 15% (a realistic threshold in real-world graphs)? Provide ablation results showing fairness metrics under varying pseudo-label error rates.

2. What specific causal assumptions (e.g., 'no unobserved confounders') underpin the causal graph construction? Provide empirical evidence (e.g., causal discovery tests) that these assumptions hold for at least one benchmark dataset (e.g., Cora, Pubmed) used in your experiments.

3. How does COSTA perform across all major fairness metrics (e.g., Equal Opportunity, Predictive Equality, Individual Fairness) on the same datasets? Show a fairness-accuracy Pareto frontier to demonstrate whether the method achieves consistent fairness improvements or trades off between metrics.

---

### Meta-Review · Area_Chair_5eTq · 2026-01-06

**Summary:**

The reviewers have the following concerns:
1. Reviewers questioned the robustness to inaccurate pseudo-labels, particularly for sensitive attributes, due to missing error propagation analysis.
2. Concerns were raised about the inconsistent and unclear definition of domain shifts, affecting experimental validity.
3. The fairness evaluation was seen as incomplete, with limited metrics and concerns about combining fairness and accuracy into a single metric.
4. Practical issues included high sensitivity to hyperparameters, missing complexity analysis, naming inconsistencies, and an inaccessible code link.

**Reviewer Concerns:**

The authors did not participate the discussion. Hence, the major concerns remain unaddressed.

**Reviewer Scores:**

The authors did not participate the discussion. Hence, I cannot foresee any changes regards the score.

---

### Decision · Program_Chairs · 2026-01-26

Reject